# Phenotypic and Genomic Characterization of *Oceanisphaera submarina* sp. nov. Isolated from the Sea of Japan Bottom Sediments

**DOI:** 10.3390/life15030378

**Published:** 2025-02-27

**Authors:** Lyudmila Romanenko, Evgeniya Bystritskaya, Nadezhda Otstavnykh, Valeriya Kurilenko, Peter Velansky, Marina Isaeva

**Affiliations:** 1G.B. Elyakov Pacific Institute of Bioorganic Chemistry, Far Eastern Branch, Russian Academy of Sciences, Prospect 100 Let Vladivostoku 159, Vladivostok 690022, Russia; ep.bystritskaya@yandex.ru (E.B.); chernysheva.nadezhda@gmail.com (N.O.); valerie@piboc.dvo.ru (V.K.); 2A.V. Zhirmunsky National Scientific Center of Marine Biology, Far Eastern Branch, Russian Academy of Sciences, Palchevskogo Street 17, Vladivostok 690041, Russia; velansky.pv@gmail.com

**Keywords:** marine bacteria, *Oceanisphaera*, phylogeny, taxonomy, pan-genome, bottom sediments, the Sea of Japan

## Abstract

A Gram-negative aerobic, motile bacterium KMM 10153^T^ was isolated from bottom sediment sampled from the Sea of Japan at a depth of 256 m, Russia. Strain KMM 10153^T^ grew in 0–12% NaCl at temperatures ranging from 4 to 42 °C and produced brown diffusible pigments. Based on the 16S rRNA gene and whole genome sequences analyses, novel bacterium KMM 10153^T^ was affiliated with the genus *Oceanisphaera* (phylum *Pseudomonadota*) showing the highest 16S rRNA gene sequence similarities of 98.94% to *Oceanisphaera arctica* KCTC 23013^T^, 98.15% to *Oceanisphaera donghaensis* BL1^T^, and similarity values of <98% to other validly described *Oceanisphaera* species. The pairwise Average Nucleotide Identity (ANI) and Average Amino Acid Identity (AAI) values between the novel strain KMM 10153^T^ and the three closest type strains *Oceanishaera arctica* KCTC 23013^T^, *Oceanisphaera litoralis* DSM 15406^T^ and *Oceanisphaera sediminis* JCM 17329^T^ were 89.4%, 89.1%, 87.41%, and 90.7%, 89.8%, 89.7%, respectively. The values of digital DNA–DNA hybridization (dDDH) were below 39.3%. The size of the KMM 10153^T^ draft genome was 3,558,569 bp, and the GC content was 57.5%. The genome of KMM 10153^T^ harbors 343 unique genes with the most abundant functional classes consisting of transcription, mobilome, amino acid metabolism, and transport. Strain KMM 10153^T^ contained Q-8 as the predominant ubiquinone and C_16:1ω7c_, C_16:0,_ and C_18:1ω7_c as the major fatty acids. The polar lipids were phosphatidylethanolamine, phosphatidylglycerol, diphosphatidylglycerol, and phosphatidic acid. Based on the distinctive phenotypic characteristics and the results of phylogenetic and genomic analyses, the marine bacterium KMM 10153^T^ could be classified as a novel *Oceanisphaera submarina* sp. nov. The type strain of the species is strain KMM 10153^T^ (=KCTC 8836^T^).

## 1. Introduction

Marine deep-sea bottom sediments are natural, unique habitats of marine biota and a valuable source of biodiversity. As important components associated with the deep-sea, sediment microorganisms play an essential role in the biochemical processes and biotic activity of marine ecosystems [1]. Studying the biodiversity of bacteria isolated from the bottom sediments of the Sea of Japan, a Gram-negative aerobic, motile bacterium KMM 10153^T^ was isolated and investigated using phenotypic and molecular methods. Based on the phylogenetic analysis, strain KMM 10153^T^ was assigned to the genus *Oceanisphaera* (phylum *Pseudomonadota*). The genus *Oceanisphaera* was described by Romanenko et al. [2] with the type species *O. litoralis*. Subsequently, it was amended by Choi et al. [3], Srinivas et al. [4], and Xu et al. [5]. Currently, the genus *Oceanisphaera* comprises nine species with a validly published and correct name and three species with an effectively but not validly published name (https://lpsn.dsmz.de/genus/oceanisphaera (accessed on 26 December 2024)) [6]. Most bacteria of the genus *Oceanisphaera* have been reported to be recovered from marine habitats, including deep-sea sediment [5], coastal sediment [2,7,8], arctic marine sediment [4], a deep-sea seamount [9], seawater of an oyster farm [3], and marine sediment collected from a cage-cultured ark clam farm [10]. Two *Oceanisphaera* species were found to be associated with the animal specimen: *Oceanisphaera avium*, which was isolated from the gut of the cinereous vulture, *Aegypius monachus* [11], and “*Oceanisphaera pacifica*”, isolated from the intestine of the beltfish, *Trichiurus japonicus* [12].

The present study defined the taxonomic position of a deep bottom sediment-derived bacterium KMM 10153^T^. Phylogenomic and pan-genome analyses of the genus *Oceanisphaera* were carried out to understand the specificity of the KMM 10153^T^ genome. A novel species, *Oceanisphaera submarina* sp. nov., is described based on combined phylogenetic analyses and phenotypic properties.

## 2. Materials and Methods

### 2.1. Bacterial Strains

Strain KMM 10153^T^ was isolated from a bottom sediment sample obtained at a depth of 256 m from the Sea of Japan (42°34.1′ N 133°34.0′ E), Russia, during the cruise 64 of the R/V Academician Oparin in June 2021. The novel bacterium was cultivated aerobically at 28 °C on marine agar 2216 (MA; BD Difco^TM^, Sparks, MD, USA) or in marine broth 2216 (MB; BD Difco^TM^, Sparks, MD, USA), tryptic soya agar (TSA) or tryptic soya broth (TSB; BD BBL^TM^, Sparks, MD, USA) and stored at −70 °C in MB 2216 supplemented with 30% (*v*/*v*) glycerol. The strain KMM 10153^T^ has been deposited in the Collection of Marine Microorganisms (KMM), Russia, and in the Korean Collection for Type Cultures (KCTC), Korea, as KCTC 8836^T^. The type strain *Oceanisphaera arctica* KCTC 23013^T^ was provided from the Korean Collection for Type Cultures to be used in the comparative phenotypic analyses.

### 2.2. Phenotypic Characterization

Gram-staining, oxidase and catalase tests, and motility (the hanging drop technique) were determined as described by Gerhardt et al. [13]. The morphology of cells grown on MA 2216 and negatively stained with a 1% phosphotungstic acid on carbon-coated 200-mesh copper grids was examined by Libra 120 FE (Carl Zeiss, Oberkochen, Germany). Hydrolysis of casein, starch, gelatin, L-tyrosine, Tweens 20, 40, and 80, as well as nitrate reduction (sulfanilic acid/α-naphthylamine test), and production of H_2_S from thiosulfate, were carried out using MA 2216, TSA or glucose-peptone media prepared based on the artificial seawater (ASW) as described in previous papers [14,15]. The ASW contained (per liter of distilled water) 24 g NaCl, 4.9 g MgCl_2_, 2.0 g MgSO_4_, 0.5 g CaCl_2_, 1.0 g KCl, 0.01 g FeSO_4_. The salinity range for growth was tested in a medium prepared with NaCl-free ASW containing different concentrations of NaCl [0–15.0% (in increments of 1.0%), *w*/*v*], 0.1% yeast extract, and 1% Bacto peptone. The pH range (pH 4.0–11 in increments of 0.5 pH units) was determined in MB and adjusted by the addition of potassium phosphate, Tris-HCl, or sodium carbonate/sodium bicarbonate buffers. The salinity and pH ranges for growth were investigated in test tubes on a shaker for 5 days, at which time the optical density was estimated at 600 nm. The temperature range and optimum values were tested with growth assessed by observation of visible colonies on MA and TSA during incubation for 5 days at different temperatures (4, 15, 22, 25, 28, 30, 35, 37, 40, 41, 42, 43, and 45 °C). The cell suspension, corresponding to McFarland standard 1 (bioMérieux, Marcy l’Etoile, France), was prepared in 0.85% NaCl, and a loopful (10 μL) of this suspension was streaked to each test plate.

Hydrolysis of DNA was examined using DNase Test Agar (BD BBL^TM^, Sparks, MD, USA). The formation of a transparent zone around a spot of bacterial cells was considered a positive result. Citrate utilization was tested on Simmons citrate agar (HiMedia Laboratories, Mumbai, India). Biochemical tests were performed using API 20E, API ID32 GN, and API ZYM test kits (bioMérieux, Marcy-l’Étoile, France) as described by the manufacturer.

### 2.3. Polar Lipids, Fatty Acids, and Quinone Analyses

Strains KMM 10153^T^, *O*. *arctica* KCTC 23013^T^, and *O. litoralis* KMM 3654^T^ were grown on TSA at 28 °C for 3 days. Lipids were extracted as described in [16], and polar lipids were analyzed using a two-dimensional thin-layer chromatograph according to [17,18]. Fatty acid methyl esters were obtained according to the MIDI method [19] and detected on a GC–2010 chromatograph (Shimadzu Corporation, Kyoto, Japan) with a flame ionization detector and a GC-MS QP2020 (Shimadzu Corporation, Kyoto, Japan) as described in previous works [14,15]. Identification of double-bond and methyl group positions in fatty acids was determined according to [20]. Isolation and analysis of quinones were carried out by HPLC according to [21] on a Shimadzu LC-30 chromatograph (Shimadzu Corporation, Kyoto, Japan) with a photodiode array detector SPD-M30A, as described in [22].

### 2.4. Phylogenetic Analysis of 16S rRNA Gene

The DNA of strain KMM 10153^T^ extracted by NucleoSpin Tissue kit (Macherey–Nagel, Düren, Germany) was used for PCR-amplification of the 16S rRNA gene as described in a previous paper [22] as well as for the DNA library preparation. The obtained sequence was used to calculate pairwise 16S rRNA gene sequence similarities using the EzBioCloud service, accessed on 14 December 2024 [23]. The 16S rRNA phylogenies were reconstructed by the GGDC web server (http://ggdc.dsmz.de/, updated on 6 May 2024) [24] using the DSMZ phylogenomics pipeline [25]. Maximum likelihood (ML) and maximum parsimony (MP) trees were inferred from the alignment with RAxML [26] and TNT [27], respectively. The robustness of phylogenetic trees was estimated by 1000 bootstrapping replicates.

### 2.5. Whole-Genome Sequencing and Genome-Based Phylogenetic Analysis

DNA library of strain KMM 10153^T^ was prepared using the DNA Flex kit (Illumina, San Diego, CA, USA) and sequenced at an Illumina MiSeq platform using 2 × 150-bp read paired-end runs. The reads were processed using Trimmomatic version 0.39 [28], and quality control was checked using FastQC version 0.11.8 (https://www.bioinformatics.babraham.ac.uk/projects/fastqc/, accessed on 10 January 2024) and de novo assembled using SPAdes version 3.15.3 [29]. The genome completeness and contamination were accessed with CheckM version 1.1.3 [30]. Gene annotation was performed with RAST [31] and NCBI PGAP [32].

Genome-based phylogeny was inferred with PhyloPhlAn version 3.0.1 [33] using 400 conserved protein sequences and an ML tree with RAxML version 8.2.12 [26]. The pan-genome of the *Oceanisphaera* type strains with metabolism analysis was carried out using the Anvi’o workflow version 8 as described at https://merenlab.org/2016/11/08/pangenomics-v2/, accessed 24 December 2024 [34]. The average pairwise values of Nucleotide Acid Identity (ANI), Amino Acid Identity (AAI), and in silico DNA–DNA hybridization (dDDH) were obtained using the ANI/AAI-Matrix server [35] and TYGS platform [36].

To predict carbohydrate-active enzymes (CAZymes), dbCAN3 meta server version 10 with default settings was used (http://cys.bios.niu.edu/dbCAN2, accessed on 10 December 2024) [37,38]. Biosynthetic gene clusters of secondary metabolites were identified and annotated using the antiSMASH server, version 7.0 (https://antismash.secondarymetabolites.org, accessed on 10 December 2024) [39]. Identification of the Secretion Systems components was conducted with MacSyFinder version 2.1.4 (TXSScan-1.1.3) [40]. The heat maps and bar plots were visualized using the pheatmap version 1.0.12 and ggplot2 version 3.5.1 packages in RStudio version RStudio/2024.09.1+394 with R version 4.4.2. Fonts and sizes in all figures were edited manually in Adobe Photoshop CC 2018 for better visualization. The functional and ecological analyses of the strain were performed using the Protologger web tool [41], https://www.protologger.de/ accessed 28 December 2024.

## 3. Results and Discussion

### 3.1. Phylogenetic and Phylogenomic Analyses

Based on the analyses of the 16S rRNA gene sequence similarity and the phylogenetic tree topology, new strain KMM 10153^T^ was assigned to the genus *Oceanisphaera* (phylum *Pseudomonadota*). It demonstrated the highest 16S rRNA gene sequence similarity of 98.94% to *O*. *arctica* KCTC 23013^T^, followed by *O. donghaensis* BL1^T^ with 98.15% and *O*. *litoralis* DSM 15406^T^ (=KMM 3654^T^) with 97.94%. Similarity values of <98% with other validly described *Oceanisphaera* species were observed. On the 16S rRNA phylogenetic tree (Figure 1), strain KMM 10153^T^ was located on a branch of the genus *Oceanisphaera* clade and clustered with two type strains, *O*. *arctica* KCTC 23013^T^ and *Oceanisphaera psychrotolerans* LAM-WHM-ZC^T^ with low bootstrap support.

On a phylogenomic tree constructed using 400 conserved translated proteins (Figure 2A), strain KMM 10153^T^ formed a separate species lineage with 99% bootstrap support. Its nearest neighbors were *O*. *arctica* KCTC 23013^T^, *O. litoralis* DSM 15406^T^, and *Oceanisphaera sediminis* JCM 17329^T^. The ANI/AAI values (Figure 2B, Appendix A) between strain KMM 10153^T^ and *O. arctica* KCTC 23013^T^, *O. litoralis* DSM 15406^T^ and *O. sediminis* JCM 17329^T^ were 89.4%/90.7%, 89.1/89.8 and 87.4/89.7%, respectively, which did not exceed the 95–96% threshold value accepted for the species differentiation [42,43,44]. The dDDH values (Figure 2A, Appendix A) between strain KMM 10153^T^ and *O. arctica* KCTC 23013^T^, *O. litoralis* DSM 15406^T,^ and *O. sediminis* JCM 17329^T^ were measured at 39.3, 37.9, and 34.2%, respectively, which are lower than the dDDH value of accepted 70% threshold [45,46]. Therefore, the genomic and phylogenetic analysis data indicate that strain KMM 10153^T^ could be classified as a separate species of the genus *Oceanisphaera*.

### 3.2. Genomic Characteristics and Pan-Genome Analysis

The draft genome of KMM 10153^T^ was de novo assembled into 35 contigs, with an N50 value of 179,799 bp (Table 1). The genome size is estimated to be 3,558,569 bp. It contains 3369 coding sequences and 78 RNAs. The total GC content of the KMM 10153^T^ genome was 57.5%, which corresponds to other related type strains (Table 1). The genome sequence completeness was 99.51%, and contamination was 1.35%. Two 16S rRNA gene sequences retrieved from the KMM 10153^T^ genome sequence were 100% identical to those amplified by PCR (OQ346283.1). The KMM 10153^T^ genome sequencing data correspond to the updated minimal standards used in current bacterial taxonomy [46,47].

To understand the specificity of the KMM 10153^T^ genome, the pan-genome analysis of the genus *Oceanisphaera* based on available genomes of type strains was performed (Figure 2B). A total of 7101 gene clusters (GCs) with 31,600 gene calls were determined, which can be organized as core, shell, and cloud using Euclidian distance and Ward ordination. The core genome of the genus *Oceanisphaera* included 1924 gene clusters of 19,518 genes. Among them, most annotated clusters assigned to COG classes (Figure 3A) represented translation (J, 11.07%), amino acid metabolism and transport (E, 6.86%), cell wall/membrane/envelope biogenesis (M, 6.44%), post-translational modification, protein turnover, chaperone functions (O, 5.51%), replication and repair (L, 5.25%), and coenzyme metabolism (H, 5.15%). The shell genome, including 7390 genes assigned to 1342 GCs, was associated with the COG classes E (6.86%) and M (5.51%) as well as general functional prediction (R, 6.11%), energy production and conversion (C, 6.04%), transcription (K, 6.04%), and inorganic ion transport and metabolism (P, 5.66%). The cloud genome consisted of 3835 GCs with 4692 gene calls, of which 43.08% of the genes were not assigned to the COG functional class (Figure 3A).

Moreover, each of the *Oceanisphaera* genomes contained from 183 to 780 unique genes (Figure 3B). The largest number of unique genes, including paralogs, was found in genomes of *O. psychrotolerans* LAM-WHM-ZC^T^ (492 genes) and *O. litoralis* DSM 15406^T^ (416), followed by *O. ostreae* CCUG 60525^T^ (371), *O. sediminis* JCM 17329^T^ (350), *O. arctica* KCTC 23013^T^ (255), “*O. pacifica*” D M8^T^ (239), and *O. avium* AMac2203^T^ (238). The genomes of *O. profunda* SM1222^T^ and *O. marina* CGMCC 1.15923^T^ accounted for the smallest number of unique genes of 228 and 209, respectively. The number of unique genes for KMM 10153^T^ was 343, with the most abundant functional classes: K, transcription; X, mobilome: prophages, transposons; E, amino acid metabolism, and transport.

A comparison of metabolic pathway completeness identified some differences between KMM 10153^T^ and other *Oceanisphaera*-type strains (Appendix A). KMM 10153^T^ demonstrated genetic capability to biosynthesize phenylalanine (M00024), tyrosine (M00025), cobalamin (M00122), and betaine (M00555). A module M00919 (ectoine degradation) was absent only in KMM 10153^T^. Interestingly, all strains except for KMM 10153^T^ and *O*. *ostreae* CCUG 60525^T^ possessed M00793 (dTDP-L-rhamnose biosynthesis).

### 3.3. In Silico Analysis of Hydrolytic and Biosynthetic Potentials

Functional characteristics of KMM 10153^T^ were additionally predicted by the Protologger server [41]. Among 3264 identified coding sequences, 235 genes were classified as transport proteins, while 83 genes were responsible for secretion. The number of unique enzymes encoded by the KMM 10153^T^ genome was 992. Strain KMM 10153^T^ was predicted to be able to produce acetate from acetyl-CoA (EC:2.3.1.8, 2.7.2.1), propionate from propanoyl-CoA (EC:2.3.1.8, 2.7.2.1), and L-glutamate from ammonia (EC:6.3.1.2, 1.4.1.-). The genome also has pathways for the biosynthesis of siroheme from glutamate (EC:6.1.1.17, 1.2.1.70, 5.4.3.8, 4.2.1.24, 2.5.1.61, 4.2.1.75, 2.1.1.107/1.3.1.76/4.99.1.4), riboflavin (vitamin B2) from GTP (EC:3.5.4.25, 3.5.4.26, 1.1.1.193, 3.1.3.104, 4.1.99.12, 2.5.1.78, 2.5.1.9, 2.7.1.26, 2.7.7.2), biotin (vitamin B7) from pimeloyl-ACP/CoA (EC:2.3.1.47, 2.6.1.62, 6.3.3.3, 2.8.1.6), and folate (vitamin B9) from 7,8-dihydrofolate (EC:1.5.1.3). Cbb3-type cytochrome C oxidase was predicted in the genome based on the presence of subunits I, II, III and IV. The 25 flagella-related genes were recognized in the genome. Additionally, a single CRISPR array was detected in the KMM 10153^T^ genome.

An ecological analysis of KMM 10153^T^ was performed based on Protologger server results. Its 16S rRNA gene sequence was matched against 19,000 amplicon datasets. The results indicated that the low distribution of KMM 10153^T^ is similar to that of the sequences in these datasets. The highest percentage of the sequences were present in the marine sediment (1.7%). According to the Protologger server results, no metagenome-assembled genomes (MAGs) matching the KMM 10153^T^ genome were identified from >49,000 MAGs obtained from 10 environments. These findings, therefore, suggest low abundance and distribution of KMM 10153^T^ in different environments (Appendix A).

The dbCAN analysis [37,38] showed that the genome of KMM 10153^T^ harbors a total of 45 carbohydrate-active enzymes (CAZymes) distributed into 22 glycosyltransferases (GTs), eight glycoside hydrolases (GHs), four carbohydrate esterases (CEs), seven auxiliary activities, and four carbohydrate-binding modules (Figure 4A). The most common GHs among *Oceanisphaera* type strains belonged to GH23 and GH103 families, involved in peptidoglycan degradation. The main *GT* families were GT2 and GT4, related to glycoprotein biosynthesis. CEs were represented mainly by CE4 enzymes, which are capable of removing acetyl groups in chitin.

Secondary metabolite biosynthesis gene clusters (BGCs) encoded by the KMM 10153^T^ and *Oceanisphaera* type strains genomes were analyzed on the antiSMASH server [39]. It was found that the KMM 10153^T^ genome contains BGCs for ectoine, hydrogen-cyanide, NI-siderophore, and RiPP-like type peptide (Figure 4B). The NI-siderophore gene cluster showed 45% similarity to the known vibrioferrin BGCs [48,49].

Eight putative transport secretion systems (TSS), crucial for global cellular functions, were detected in the *Oceanisphaera* genomes (Figure 4C) using the MacSyFinder version 2 [40]. All *Oceanisphaera* strains shared genes for the type I secretion system (T1SS) and the mannose-sensitive hemagglutinin pilus (MSH, type 4 pili family). The bacterial flagellum, related to the type 3 secretion system (T3SS), was also predicted in most *Oceanisphaera* strains except *O*. *arctica* KCTC 23013^T^ and *O. psychrotolerans* LAM-WHM-ZC^T^. It is interesting to note that KMM 10153^T^ had the maximum number of genes related to the Tad (tight adherence) pilus, allowing the attachment on the diatom surfaces [50].

Thus, unique properties of KMM 10153^T^ predicted on its genome include the syntheses of betaine from choline, cobalamin (B12) from cobyrinate a,c-diamide as well as vibrioferrin-like siderophore production and a CRISPR array. These features may provide survival benefits and adaptation in specific ecological niches.

### 3.4. Morphological, Physiological, and Biochemical Characteristics of Strain KMM 10153^T^

Strain KMM 10153^T^ represented spherical or ovoid bacterial cells, 1.6–2.0 μm long and 0.9–1.1 μm in diameter, motile with one to three polar and/or lateral flagella (Appendix A).

The novel bacterium KMM 10153^T^ was able to grow in the range of 0–12% NaCl and at a temperature interval of 4–42 °C. Strain KMM 10153^T^ assimilated sodium acetate, L-alanine, glycogen, L-serine, D-mannitol, D-glucose (weakly), L-histidine, L-proline, 3-hydroxybutyric acid of carbon sources in the 32ID GN tests. It could not degrade gelatin, casein, starch, DNA, and Tweens 40 and 80 (Table 2). Other cultural, physiological, and biochemical characteristics of the novel bacterium are given in Table 2 and in the species description.

### 3.5. Chemotaxonomic Characteristics of Strain KMM 10153^T^

Strain KMM 10153^T^ contained Q-8 as the predominant ubiquinone and C_16:1ω7c_, C_16:0_, and C_18:1ω9c_ as the major fatty acids. Its fatty acid profile obtained was similar to those of related type strains *O. arctica* KCTC 23013^T^ and *O. litoralis* DSM 15406^T^ (Table 3). The polar lipids of the strain KMM 10153^T^ comprised phosphatidylethanolamine (PE), phosphatidylglycerol (PG), diphosphatidylglycerol (DPG), and phosphatidic acid (PA) (Appendix A). The substances PE, PG, DPG, and PA were also found in the related type strains *O. arctica* KCTC 23013^T^ and *O. litoralis* DSM 15406^T^. The DNA GC content of 57.5% was calculated from the genome sequence of the strain KMM 10153^T^. Chemotaxonomic characteristics of the strain KMM 10153^T,^ including ubiquinone Q-8, the predominance of C_16:1ω7c_, C_16:0_, and C_18:1ω9c_, the major polar lipid components of PE, PG, PC, and the DNA GC content, are in line with those previously described for *Oceanisphaera* species [3,4,5] justifying its placement to this genus.

## 4. Conclusions

In summary, the phylogenetic and genetic distinctions obtained for the novel isolate KMM 10153^T^ were additionally supported by phenotypic differences, including the ability to form polar and lateral flagella, its growth temperature and salinity ranges, enzyme activity, carbon source assimilation, and substrate hydrolysis (Table 1). Based on the combined phylogenetic and phylogenomic evidence and phenotypic characteristics, it is proposed to classify strain KMM 10153^T^ as a novel species, *Oceanisphaera submarina* sp. nov.

### Description of Oceanisphaera submarina sp. nov.

*Oceanisphaera submarina* (sub.ma.ri’na. L. prep. sub, under; L. fem. adj. *marina*, of the sea, marine; N.L. fem. adj. *submarina*, from under the sea).

Aerobic, Gram-negative, catalase- and oxidase-positive, spherical or ovoid bacterial cells, 1.6–2.0 μm long and 0.9–1.1 μm in diameter, motile with one to three polar and/or lateral flagella. On TSA, formed light brown hemi-translucent shiny colonies with regular edges of 3–4 mm in diameter, producing brown diffusible pigments into the medium. Growth occurs in 0–12% NaCl (*w*/*v*) and is optimal in 3–4% NaCl. The temperature range for growth was 4–42 °C, with an optimum of 28–30 °C. Weak growth was observed at 12% NaCl and at 42 °C. The pH range for growth is 5.0–10.5, with an optimum of 6.5−7.5. Positive for hydrolysis of Tween 20, nitrate reduction, and H_2_S production from thiosulfate. Negative for hydrolysis of casein, gelatin, DNA, Tween 40, Tween 80, starch, and citrate utilization (Simmon’s citrate agar test). On an L-tyrosine-containing medium, brown pigments are produced, but a clearance zone is not formed.

In the API 20E negative for ONPG, arginine dihydrolase, lysine decarboxylase, ornithine decarboxylase, citrate utilization, H_2_S, and urease production under anaerobic conditions, tryptophane deaminase, indole production, acetoin production (Voges-Proskauer reaction), gelatin hydrolysis, and oxidation/fermentation of D-sucrose, D-glucose, D-mannitol, inositol, D-sorbitol, L-rhamnose, D-melibiose, amygdalin, and L-arabinose.

In the ID32 GN tests, positive for the assimilation of sodium acetate, L-alanine, glycogen, L-serine, D-mannitol, D-glucose (weakly), L-histidine, L-proline, 3-hydroxybutyric acid; and negative for the assimilation of L-rhamnose, N-acetylglucosamine, D-ribose, inositol, D-sucrose, D-maltose, itaconic acid, suberic acid, sodium malonate, lactic acid, potassium 5-ketogluconate, 3-hydroxybenzoic acid, salicin, D-melibiose, L-fucose, D-sorbitol, L-arabinose, propionic acid, capric acid, trisodium citrate, potassium 2-ketogluconate, and 4-hydroxybenzoic acid.

Enzymatic activity tests were positive for alkaline phosphatase, esterase (C 4), esterase lipase (C 8), leucine arylamidase, naphthol-AS-BI-phosphohydrolase; and negative for lipase (C 14), valine arylamidase, cystine arylamidase, trypsin, α-chymotrypsin, acid phosphatase, α-galactosidase, β-galactosidase, α-glucosidase, β-glucosidase, β-glucuronidase, N-acetyl-β-glucosaminidase α-mannosidase, and α-fucosidase.

The dominant respiratory quinone was ubiquinone Q-8. Major fatty acids were C_16:1ω7c_, C_16:0_, and C_18:1ω9c_. The polar lipids consisted of phosphatidylethanolamine, phosphatidylglycerol, diphosphatidylglycerol, and phosphatidic acid. The DNA GC content of 57.5% was calculated from the genome sequence.

The type strain of the species is KMM 10153^T^ (=KCTC 8836^T^), isolated from a bottom sediments sample collected from the Sea of Japan, Russia.

The DDBJ/GenBank accession number for the 16S rRNA gene sequence of strain KMM 10153^T^ is OQ346283.1.

The annotated draft genome of type strain KMM 10153^T^ comprising 3,558,569 bp is deposited in the NCBI GenBank database under the accession number JBKGFN000000000.

## Figures and Tables

**Figure 1 life-15-00378-f001:**
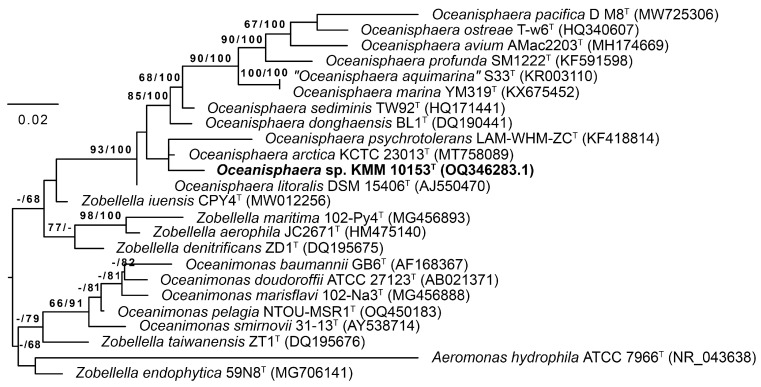
Maximum Likelihood/Minimal Parsimony (ML/MP) 16S rRNA tree showing the position of the new strain KMM 10153^T^ (in bold) among type strains of the genera *Oceanisphaera*, *Zobellella*, and *Oceanimonas*. The ML tree was inferred under the GTR + GAMMA model. The numbers (ML/MP) show bootstrap values greater than 60% measured with 1000 replicates. *Aeromonas hydrophila* was used as an outgroup taxon. The bar shows 0.02 substitutions per nucleotide position.

**Figure 2 life-15-00378-f002:**
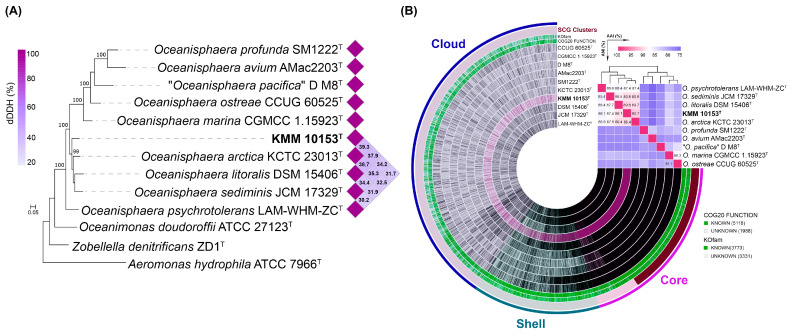
(**A**) ML genomic tree presenting a phylogenetic position of the new strain KMM 10153^T^ among *Oceanisphaera* type strains based on concatenated sequences of 400 translated proteins. The ML tree is reconstructed under the LG + Γ model with 100 non-parametric bootstrapping replicates. Bar, 0.05 substitutions per amino acid position. The dDDH values of the KMM 10153^T^ clade are shown as a heatmap. (**B**) Pan-genome of KMM 10153^T^ and nine type strains of the genus *Oceanisphaera*. The circle bars represent the presence/absence of 7101 pan-genomic clusters in each genome. The heatmap placed in the upper right corner shows pairwise ANI and AAI values in percentages.

**Figure 3 life-15-00378-f003:**
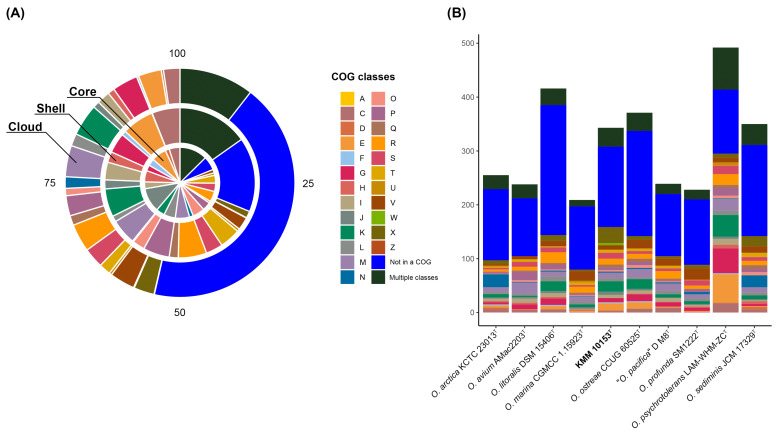
The COG class distribution predicted in the core, shell, and cloud genomes of the genus *Oceanisphaera* in percentages (**A**) and the number of unique genes assigned to COG functional classes among *Oceanisphaera* type strains (**B**). Classes are as follows: A, RNA processing and modification; C, energy production and conversion; D, cell cycle control and mitosis; E, amino acid metabolism and transport; F, nucleotide metabolism and transport; G, carbohydrate metabolism and transport; H, coenzyme metabolism, I, lipid metabolism; J, translation; K, transcription; L, replication and repair; M, cell wall/membrane/envelope biogenesis; N, cell motility; O, post-translational modification, protein turnover, chaperone functions; P, inorganic ion transport and metabolism; Q, secondary structure; R, general functional prediction only; S, function unknown; T, signal transduction; U, intracellular trafficking and secretion; V, defense mechanisms; W, extracellular structures; X, mobilome: prophages, transposons; Z, cytoskeleton; Multiple classes, genes assigned to two or more COG categories; Not in a COG, COG not defined.

**Figure 4 life-15-00378-f004:**
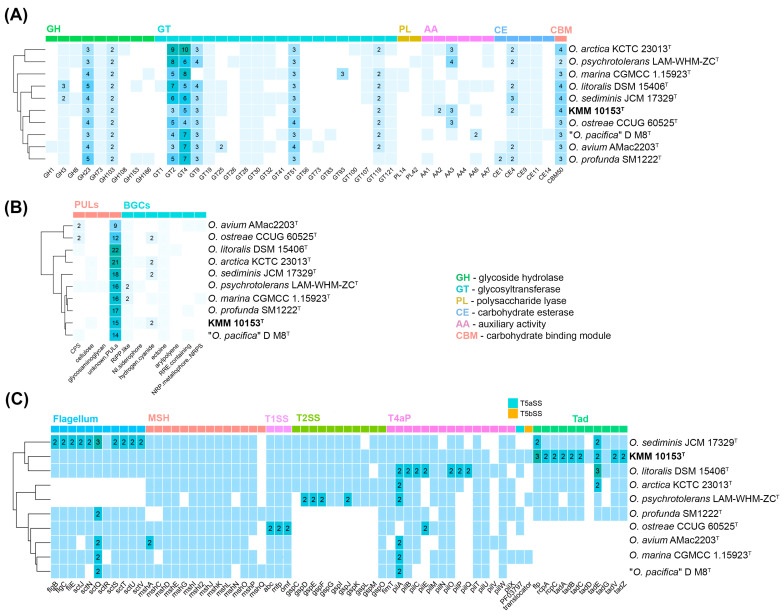
The distribution of CAZymes, PULs, biosynthetic and secretion system gene clusters in KMM 10153^T^, and type strains of the *Oceanisphaera* genus. (**A**) Heatmap of the CAZyme family abundance in *Oceanisphaera* species. (**B**) Heatmap of the secretion system gene clusters in *Oceanisphaera* species. (**C**) Heatmap of the PULs and biosynthetic gene clusters in *Oceanisphaera* species.

**Table 1 life-15-00378-t001:** Genomic features of KMM 10153^T^ and related *Oceanisphaera* type strains.

Feature	1	2	3	4	5
Assembly level	contig	scaffold	scaffold	scaffold	contig
Genome size (Mb)	3.6	3.6	3.8	3.4	3.8
Number of contigs	35	38	68	68	70
G + C Content (mol%)	57.5	57.5	57.5	58.5	58.5
N50 (Kb)	179.8	221.3	136.7	105.4	106.6
L50	7	8	11	11	11
Coverage	92×	100×	526×	337×	20×
Protein coding genes	3369	3214	3459	3149	3066
rRNAs (5S/16S/23S)	2/2/2	4/2/1	0/0/8	1/3/2	1/2/2
tRNAs	72	78	95	81	46
checkM completeness (%)	99.51	99.05	98.44	98.30	89.64
checkM contamination (%)	1.35	2.03	0.85	0.97	2.03
WGS project	JBKGFN000000000	JBHLZJ01	BAABDS01	JAFBBE01	MDKE01
Genome assembly	ASM4652130v1	ASM4243140v1	ASM3953974v1	ASM1690701v1	ASM187048v1

Strains: 1, KMM 10153^T^; 2, *O. arctica* KCTC 23013^T^; 3, *O. sediminis* JCM 17329^T^; 4, *O. litoralis* DSM 15406^T^; 5, *O. psychrotolerans* LAM-WHM-ZC^T^.

**Table 2 life-15-00378-t002:** Differential characteristics of KMM 10153^T^ and the type strains of related *Oceanisphaera* species.

Characteristic	1	2	3	4
DNA GC content (%) *	57.5	58.5	57.5	55.5
Motility	+	+	−	−
Growth in NaCl:				
0%	+	−	+	+
10%	+	+	−	+
12%	(+)	−	−	−
Growth at:				
40 °C	+	+	−	+
42 °C	(+)	+	−	−
Hydrolysis of:				
DNA	−	−	+	ND
Tween 80	−	−	−	+
Nitrate reduction	+	+	−	+
Urease	−	+	+	(+)
Assimilation of:				
D-mannitol	+	−	−	ND
L-rhamnose	−	−	−	+
L-arabinose	−	−	−	+
L-fucose	−	−	−	+
4-hydroxybenzoic acid	−	+	−	ND
L-proline	+	−	+	ND
L-serine	+	+	+	−
Enzyme activity (API ZYM):				
Alkaline phosphatase	+	(+)	+	+
Esterase lipase C 4	+	−	+	+
Valine arylamidase	−	(+)	−	(+)
Cystine arylamidase	−	−	−	(+)
Trypsin	−	−	−	(+)
β-glucuronidase	−	+	−	−

Strains are as follows: 1, KMM 10153^T^; 2, *O. litoralis* KMM 3654^T^; 3, *O. arctica* KCTC 23013^T^ (data were obtained from present study unless otherwise indicated); 4, *O. psychrotolerans* LAM-WHM-ZC^T^ (data from Zhou et al., 2015 [8]). +, Positive; −, negative; (+), weak reaction; ND, not determined. All strains were positive for catalase, oxidase, and growth at 4 °C; in the API ZYM tests were positive for esterase lipase C8, leucine arylamidase, naphthol phosphohydrolase and negative for indole production, D-glucose fermentation, arginine dihydrolase, hydrolysis of gelatin, α-glucosidase. All strains (except *O. psychrotolerans* LAM-WHM-ZC^T^ for which no data are available) were positive for assimilation of L-alanine, L-histidine, 3-hydroxybutyric acid; negative for hydrolysis of casein, starch, Tween 40; lipase C14, α-chymotrypsin, acid phosphatase, α-galactosidase, β-galactosidase, β-glucosidase, N-acetyl-β-glucosaminidase, α-mannosidase, and α-fucosidase. * DNA GC contents of strains KMM 10153^T^, *O. litoralis* KMM 3654^T^, and *O. arctica* KCTC 23013^T^ and *O. psychrotolerans* LAM-WHM-ZC^T^ were derived from their genome assembly data (Table 1).

**Table 3 life-15-00378-t003:** Cellular fatty acid composition (%) of the new KMM 10153^T^ and the related type strains of the genus *Oceanisphaera*.

Fatty Acid	1	2	3	4
C_12:0_	5.7	6.6	7.9	16.2
C_14:0_	-	-	-	1.6
C_15:0_	Tr	Tr	1.4	-
C_16:0_	24.0	16.6	13.0	15.9
C_17:0_	1.1	Tr	1.0	2.0
C_18:0_	2.6	Tr	Tr	2.8
C_16:1ω7*c*_	33.5	40.3	41.6	24.5 *
C_16:1ω5*c*_	1.3	Tr	Tr	-
C_17:1ω8*c*_	Tr	1.0	1.3	0.9
C_18:1ω7*c*_	17.5	20.3	16.6	9.9 **
iso-C_17:0_	-	-	-	2.4
C_14:0_ 3-OH	7.5	8.0	9.2	-

Strains are as follows: 1, KMM 10153^T^; 2, *O. litoralis* KMM 3654^T^; 3, *O. arctica* KCTC 23013^T^ (data were obtained from the present study); 4, *O. psychrotolerans* LAM-WHM-ZC^T^ [8]. Fatty acids representing <1% in all strains tested were not shown; Tr, trace amounts (<1%); -, not detected. * Summed C_16:1ω7c_/C_16:1ω6c_; ** Summed C_18:1ω7c_/C_18:1ω6c_.

## Data Availability

The strain type of the species *Oceanisphaera submarina* sp. nov. is strain KMM 10153^T^ (=KCTC 8836^T^) isolated from a bottom sediments sample collected from the Sea of Japan, Russia. The DDBJ/GenBank accession numbers for the 16S rRNA gene and genome sequences of strain KMM 10153^T^ are OQ346283.1 and JBKGFN000000000, respectively.

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
