# Peer review of "Phenotypic and Genomic Characterization of *Oceanisphaera submarina* sp. nov. Isolated from the Sea of Japan Bottom Sediments"

_life, 2025, doi:10.3390/life15030378_

Round 1

Reviewer 1 Report

Comments and Suggestions for Authors

The manuscript concerns Phenotypic and genomic characterization of Oceanisphaera submarina sp. nov., isolated from the Sea of ​​Japan bottom sediments. It is interesting, but contains numerous errors that should be rejected.

1. The percentage of plagiarism is of great importance. It is very high. The authors should carefully analyze the manuscript and then correct it so that it does not contain content borrowed from other publications (42%). It is also noted that 15% of the text was borrowed from the mdpi platform.

2. Strain KMM 10153T was isolated from a bottom sediment sample obtained at a depth of 66 of 256 m from the Sea of ​​Japan, Russia. No in-depth analysis of the sea, sources of pollution, their impact on bacteria, water pH, maps, etc.

3. No information on whether the bacteria were cultured according to specific standards (which ones), or whether the experiment was invented by the authors...

4. Hydrolysis of DNA was examined using DNase Test. No description of the test.

5. No specific summary.

Comments on the Quality of English Language

Should be corrected

Author Response

Responses to Reviewer 1.

Comment 1: The manuscript concerns Phenotypic and genomic characterization of Oceanisphaera submarina sp. nov., isolated from the Sea of Japan bottom sediments. It is interesting, but contains numerous errors that should be rejected.

Response 1: Thank you very much for taking the time to review our manuscript. Overall, we have tried to follow your comments to improve our manuscript.

Comment 2: The percentage of plagiarism is of great importance. It is very high. The authors should carefully analyze the manuscript and then correct it so that it does not contain content borrowed from other publications (42%). It is also noted that 15% of the text was borrowed from the mdpi platform.

Response 2: Based on your comment, we checked our text with two available plagiarism detection programs. They were both able to describe most of the text as unique, except for the Materials and Methods section, which indicated our earlier work. This applies to very short phrases that we cannot even call them as “short direct quotes.” We would be grateful if you would kindly provide your version of checking our manuscript for plagiarism.

Comment 3: Strain KMM 10153T was isolated from a bottom sediment sample obtained at a depth of 66 of 256 m from the Sea of Japan, Russia. No in-depth analysis of the sea, sources of pollution, their impact on bacteria, water pH, maps, etc.

Response 3. We included coordinates (N 42° 34.1´ E 133° 34.0´) of the sediment sampling site (Line 68). Unfortunately, parameters such as sources of pollution, their impact on bacteria, water pH were not evaluated during this expedition.

Comment 4: No information on whether the bacteria were cultured according to specific standards (which ones), or whether the experiment was invented by the authors.

Response 4: As indicated in the Materials and Methods section the bacteria tested were cultivated according to standard procedures (Ref. 13. Gerhardt et al. 1994 Methods for General and Molecular Bacteriology), using commercial nutrient media for their growth.

Comment 5: Hydrolysis of DNA was examined using DNase Test. No description of the test.

Response 5: In the text of our manuscript, we reported that DNA hydrolysis was studied using DNase Test agar (BD BBLTM). We extended this following information: “The formation of a transparent zone around a spot of bacterial cells is considered a positive result” (Lines 102-103).

Comment 6: No specific summary.

Response 6: In a taxonomic article, the conclusion corresponds to the detailed protologue (a set of associated elements representing the first publication of a new taxon). Therefore, this type of conclusion, this type of summary, is a mandatory part and the basis for validating the publication of the taxon name.

Reviewer 2 Report

Comments and Suggestions for Authors

The experimental framework should be more complete and rational since missing some of the corresponding experiments.

1. the morphological pictures of the bacteria are not given in the paper, only textual descriptions are given.

2. the last description of the optimal conditions for the growth of salinity, temperature, etc. How can you deduce them? I mean the lack of experiments.

3. the illustration of genomic characteristics could be expanded

Author Response

Responses to Reviewer 2.

The experimental framework should be more complete and rational since missing some of the corresponding experiments.

Comment 1: the morphological pictures of the bacteria are not given in the paper, only textual descriptions are given.

Response 1: Thank you very much for taking the time to review our manuscript. The transmission electron micrograph of strain KMM 10153T was not missing and has been placed in the Supplementary data file as Fig. S3.

Comment 2: the last description of the optimal conditions for the growth of salinity, temperature, etc. How can you deduce them? I mean the lack of experiments.

Response 2: These experiments were carried out as described in detail in our previous articles (Ref. 14 and 15). Here we added them to the Methods section, Lines 90-101.

Comment 3: the illustration of genomic characteristics could be expanded.

Response 3: If we understand you correctly, we need to represent the structure of the bacterial genome in the form of a diagram. Unfortunately, due to the lack of a chromosomal assembly level, such an illustration will not be sufficiently informative.

Reviewer 3 Report

Comments and Suggestions for Authors

Comments

The study characterized a novel species, named Oceanisphaera submarina sp. nov., isolated from the Sea of Japan bottom sediments. The manuscript is well-structured but requires minor revisions for clarity and consistency.

In abstract,line22-25, ” The Average Nucleotide Identity/ Average Amino Acid Identity and digital DNA-DNA hybridization values between strain КMM 10153T and O.arctica KCTC 23013T, Oceanisphaera litoralis DSM 15406T and Oceanisphaera sediminis JCM 17329T were 89.39%/ 90.66%, 89.06/ 89.79 and 87.41/ 89.72%, and 39.3, 37.9 and 34.2%, respectively.” The sentence is confusing.

For novel species, it is standard practice to provide two deposit certificates from different countries to ensure the availability of the strain for future research. Please ensure that these certificates are included and properly referenced in the manuscript.

Figure 1. ML/MP 16S rRNA tree— The full name should be clearly stated for clarity.

Based on the 16s rDNA similarity and whole genome comparing, the differential characteristics of КММ 10153T and type strains of related Oceanisphaera species, should include O. arctica KCTC 23013T,   O. donghaensis BL1T   O. litoralis DSM 15406T   and Oceanisphaera sediminis JCM 17329T. Pls revise Tables 2, 3, and 4.

Regarding strain names: Is Oceanisphaera litoralis KMM 3654T the same as Oceanisphaera litoralis DSM 15406T? Please ensure that strain names are consistent throughout the text to avoid confusion.

Comments on the Quality of English Language

The language of the manuscript should be improved.

Author Response

Comment: The study characterized a novel species, named Oceanisphaera submarina sp. nov., isolated from the Sea of Japan bottom sediments. The manuscript is well-structured but requires minor revisions for clarity and consistency.

Response: Thank you very much for taking the time to review our manuscript. We have tried to follow your comments to improve our manuscript.

Comment 1: In abstract, line 22-25, ” The Average Nucleotide Identity/ Average Amino Acid Identity and digital DNA-DNA hybridization values between strain КMM 10153T and O.arctica KCTC 23013T, Oceanisphaera litoralis DSM 15406T and Oceanisphaera sediminis JCM 17329T were 89.39%/ 90.66%, 89.06/ 89.79 and 87.41/ 89.72%, and 39.3, 37.9 and 34.2%, respectively.” The sentence is confusing.

Response 1: This sentence was re-written (Lines 21-26).

Comment 2: For novel species, it is standard practice to provide two deposit certificates from different countries to ensure the availability of the strain for future research. Please ensure that these certificates are included and properly referenced in the manuscript.

Response 2: The deposit certificate from Collection of Marine Microorganisms (KMM) is added to the Supplementary.

Comment 3: Figure 1. ML/MP 16S rRNA tree— The full name should be clearly stated for clarity.

Response 3: Abbreviated names are explained in the title of Figure 1.

Comment 4: Based on the 16s rDNA similarity and whole genome comparing, the differential characteristics of КММ 10153T and type strains of related Oceanisphaera species, should include O. arctica KCTC 23013T, O. donghaensis BL1T, O. litoralis DSM 15406T and Oceanisphaera sediminis JCM 17329T. Pls revise Tables 2, 3, and 4.

Response 4: We agree with this comment. However, the presented differential characteristics of Oceanisphaera donghaensis BL1T and Oceanisphaera sediminis JCM 17329T were insufficient for comparison due to these two species were published long ago. Instead, we have included data provided for Oceanisphaera psychrotolerans LAM-WHM-ZCT as the species that encloses the genus Oceanisphaera. Please, see tables 1, 2 and 3, columns 4 and 5. There was no Table 4.

Comment 5:

Regarding strain names: Is Oceanisphaera litoralis KMM 3654T the same as Oceanisphaera litoralis DSM 15406T? Please ensure that strain names are consistent throughout the text to avoid confusion.

Response 5: There is no mistake there. The type strain of the species Oceanisphaera litoralis has different names in different collections (KMM 3654T and DSM 15406T). Strain KMM 10153T was used to study the morphological, physiological and biochemical properties, while DSM 15406T was used to sequence its genome.

Reviewer 4 Report

Comments and Suggestions for Authors

This is a typical genome sequencing article, and it is speculated that it has undergone a round of review based on the traces of modifications. I think it is a relatively complete genome article, with only one suggestion to add the ecological significance and research value of the bacterium Oceanisphaera submarina 10153. Although there is nothing particularly special about it, it is still valuable for further research as a genome database, and I recommend publishing it.

Author Response

Comment: This is a typical genome sequencing article, and it is speculated that it has undergone a round of review based on the traces of modifications. I think it is a relatively complete genome article, with only one suggestion to add the ecological significance and research value of the bacterium Oceanisphaera submarina 10153. Although there is nothing particularly special about it, it is still valuable for further research as a genome database, and I recommend publishing it.

Response: Thank you very much for taking the time to review our manuscript. Thank you for your positive feedback on our manuscript and your recommendation to include information about the possible ecological significance of Oceanisphaera submarina 10153T. Please, Lines 299-302.

Round 2

Reviewer 1 Report

Comments and Suggestions for Authors

The percentage of plagiarism is of great importance. It is very high. The authors should carefully analyze the manuscript and then correct it so that it does not contain content borrowed from other publications (46%). It is also noted that 13% of the text was borrowed from the mdpi platform.

 Strain KMM 10153T was isolated from a bottom sediment sample obtained at a depth of 66 of 256 m from the Sea of Japan, Russia. No in-depth analysis of the sea, sources of pollution, their impact on bacteria, water pH, maps, etc. parameters such as sources of pollution, their impact on bacteria, water pH were not evaluated during this expedition.

Comments on the Quality of English Language

should be corrected

Author Response

Responses to Reviewer 1.

Comment 1: The percentage of plagiarism is of great importance. It is very high. The authors should carefully analyze the manuscript and then correct it so that it does not contain content borrowed from other publications (46%). It is also noted that 13% of the text was borrowed from the mdpi platform.

Response 1: Based on the iThenticate Report, we reduced the similarity identified in the Materials and Methods section.

Comment 2: Strain KMM 10153T was isolated from a bottom sediment sample obtained at a depth of 66 of 256 m from the Sea of Japan, Russia. No in-depth analysis of the sea, sources of pollution, their impact on bacteria, water pH, maps, etc. parameters such as sources of pollution, their impact on bacteria, water pH were not evaluated during this expedition.

Response 2. Measuring these conditions requires appropriate equipment and staff. Unfortunately, the objectives of this expedition were biological sampling and not any ecological or oceanographic tasks. Here we indicated the coordinates, depth and time of sampling (Lines 68-70).

Reviewer 2 Report

Comments and Suggestions for Authors

The author has revised the paper according to my manuscript. I am pleased to recommend to accept its publication.

Author Response

Response to Reviewer 2.

Comment: The author has revised the paper according to my manuscript. I am pleased to recommend to accept its publication.

Response: Thank you very much for taking the time to review our manuscript and for your recommendation for the acceptance of our manuscript.
